systems biology, neuroscience, cognition

peripheral vision, motion, retina, perimetry, *ora serrata*

**Author for correspondence:**
J. D. Mollon
e-mail: jm123@cam.ac.uk

# 'The last channel': vision at the temporal margin of the field

P. Veto[1], P. B. M. Thomas[2], P. Alexander[3], T. A. Wemyss[1] and J. D. Mollon[1]

[1]Department of Psychology, University of Cambridge, Downing Street, Cambridge CB2 3EB, UK
[2]NIHR Biomedical Research Centre at Moorfields Eye Hospital and UCL Institute of Ophthalmology, London EC1V 9EL, UK
[3]Department of Ophthalmology, Addenbrooke's Hospital, Hills Road, Cambridge CB2 0QQ, UK

PV, 0000-0003-3545-0098; TAW, 0000-0002-9762-5407; JDM, 0000-0001-8533-033X

The human visual field, on the temporal side, extends to at least 90° from the line of sight. Using a two-alternative forced-choice procedure in which observers are asked to report the direction of motion of a Gabor patch, and taking precautions to exclude unconscious eye movements in the direction of the stimulus, we show that the limiting eccentricity of image-forming vision can be established with precision. There are large, but reliable, individual differences in the limiting eccentricity. The limiting eccentricity exhibits a dependence on log contrast; but it is not reduced when the modulation visible to the rods is attenuated, a result compatible with the histological evidence that the outermost part of the retina exhibits a high density of cones. Our working hypothesis is that only one type of neural channel is present in the far periphery of the retina, a channel that responds to temporally modulated stimuli of low spatial frequency and that is directionally selective.

## 1. Introduction

In the foveal region, at the centre of the retina, the signals from the rods and cones are analysed by as many as 25 'pre-processors'—morphologically distinct types of ganglion cell that extract different properties of the retinal image [1–3]. These independent neural channels deliver their signals in parallel to different destinations within the visual system [4]. Histologically, this diversity of functional pathways is reflected in the presence of up to 8 closely packed layers of ganglion cells in the foveal and parafoveal regions of the retina [5].

By contrast, in the far periphery of the retina (corresponding to the outer edge of the visual field), there is only a single layer of ganglion cells. Moreover, the ganglion cells are here 'separated by long gaps, being usually grouped in twos and threes' [5]. The hypothesis that we explore in this paper is that only one functional neural channel survives in the far periphery of the retina—a channel that responds to motion. There may be different subchannels for different directions.

### (a) Perceptual properties of vision at the margin of the field

Our vision at the extreme margin of the temporal field (corresponding to the extreme nasal retina) has rather seldom been studied; but certainly, our sensitivity in this region is very primitive, in that only moving stimuli of low spatial frequency are visible [6]—an observation made originally by Exner in 1875 [7]. Yet, this region of the field is functionally important: it serves as a sentinel, alerting us to sudden movement or flicker and triggering foveation [8,9]. In the course of evolution, it has allowed our ancestors to detect predators approaching from behind; and in the modern world, it alerts us to vehicles overtaking on our offside. It also has a critical role in the monitoring of self-motion [10] and the maintenance of head and body orientation. It is known that a particular region of limbic cortex—*Area Prostriata*—is specialized for processing signals from the far periphery and is sensitive to high velocities [11,12].

## (b) Establishing the functional margin of the field

In the case of healthy observers, using natural pupils, the temporal margin of vision is reported to lie at least 90° from the line of sight [13–15], although any estimate must depend on the stimulus used [6] and although—as we show here and as is suggested by previous studies [6,8,14]—there are reliable individual differences. At 90° from the line of sight, an observer has no detailed perception of pattern and only a vague sensation of the stimulus presented [16]. If a simple light on a dark field is used, he or she may confuse cues from scattered light with those arising from direct perception; and owing to the vagueness of the percepts, it is difficult for the observer to maintain a consistent criterion for reporting the presence of a stimulus. There does not seem to exist an agreed protocol for quantifying, in a controlled and reproducible manner, the extreme temporal limit of the field, either in healthy observers or in patients [17]. In Experiment 1, we set out to establish such a protocol.

Owing to the indistinctness of stimuli in the far periphery, we need a target that gives a firm threshold between eccentricities at which it is visible and those at which it is not. This is not straightforward, since previous psychophysical work suggests that the optimal stimulus for the far periphery is a moving stimulus of very low spatial frequency (approx. 0.3 cycles per degree of visual angle) aligned with one of the cardinal axes [6]; and yet an extended moving target would not allow us precisely to localize the margin of the field. We therefore used a Gabor stimulus [18]—i.e. a sinusoidal waveform multiplied by a Gaussian function—and we moved its grating component within a Gaussian envelope that was fixed in position. It is also important that the stimulus should be of relatively high luminance: owing to the reduction in the effective area of the pupil, there is a severe attenuation in the effective luminance of a stimulus that falls in the region of 90° from the line of sight [19,20]. Jay [21] estimated the effective area of the average natural pupil at 95° to be 20% of its frontal value; and at 100°, the value was 11%.

Further, it is desirable to minimize the effects of the observer's criterion. Under the indistinct conditions of extreme peripheral vision, it is difficult for an untrained observer, or a patient in a clinic, to judge with certainty whether or not a stimulus is present. We therefore asked the observer not to report the presence or absence of the Gabor but to make a forced-choice judgement of its direction of motion.

In the case of any proposed measure of individual differences or of changes in a patient's condition, it is essential to establish that differences between observers, or changes in the same patient, are reliable ones and do not simply reflect instrumental noise or time-varying sources of variance [22]. In Experiment 1, therefore, we tested each participant twice, at an interval of at least 5 days.

When these precautions were taken, we found that it was possible to define the margin of the field with some precision for any given observer and that there were reliable individual differences between observers.

## 2. Experiment 1: methods

### (a) Participants

Twenty healthy participants were recruited for the study (12 female). Their mean age was 27.9 years (s.d. = 10.4).

All were right-handed. One further participant was unavailable for the second session and is not included in the analysis. Since the temples of eye glasses or the edges of contact lenses might interfere with vision in the far periphery, and since the targets were of very low spatial frequency, participants did not wear corrections during the experiment [17,23]. Participants received £15 compensation for taking part.

### (b) Apparatus and stimuli

We measured the temporal margin of the visual field along the horizontal meridian for the left eye of each participant (figure 1). A liquid-crystal display (LCD) computer screen (Iiyama TXA 3813MT, 1024 × 768 at 75 Hz) served as a secondary screen for fixation. A chin rest was mounted so that the observer's left eye was aligned to the centre of the fixation screen, at a distance of 45 cm. The primary screen, on which the target was presented, was a CRT (cathode-ray tube; Sony FD Trinitron GDM-F400T9, 1600 × 1200 at 85 Hz), placed 60 cm from the measured eye. The screen was γ-corrected using a 'ColorCal 2' photodiode (Cambridge Research Systems, Rochester, UK).

Measurements took place in a dark room. The table for the apparatus had a matte black surface and no objects were placed around the stimulus screen, to avoid possible reflections of the stimulus. The gaze position and pupil size were recorded from both eyes at 30 Hz (LiveTrack Fixation Monitor, Cambridge Research Systems). The right eye's view of the fixation screen was blocked by a flap, mounted on the forehead bar of the chin rest. This allowed simultaneous tracking of both eyes, since the eye-tracker was positioned below the fixation screen.

The fixation point (white during stimulus presentation, red between trials; diameter = 0.18°) was presented on the fixation screen, at horizontal locations ±15° from the centre. By moving the fixation point rather than the test stimulus from trial to trial, we avoided any possible instrumental variation in the test as we varied eccentricity.

The test stimulus was an achromatic Gabor patch ($\sigma = 0.75°$; cut-off at $3.5 \times \sigma$ from the centre). Its mean luminance was 47 cd.deg$^{-2}$ and its chromaticity, in terms of the 1931 standard observer of the Commission Internationale de l'Éclairage (CIE), was $x, y = 0.268, 0.299$. The surround was black. The test stimulus was presented at a right angle to the line of sight when the participant looked straight ahead (i.e. at the centre of the fixation screen). While the location of the Gaussian patch was fixed, the sinusoidal component of the Gabor (a vertical grating of 0.55 cycle.deg$^{-1}$) travelled horizontally in either direction at 3° s$^{-1}$. The spatial frequency chosen for the Gabor followed that chosen by To *et al.* (see [6, p. 208]) and was a compromise between two opposing requirements: a frequency high enough to minimize the modulation of overall light flux as phase changed and low enough to be close to the maximal sensitivity in the far periphery. The starting phase of the grating was random from trial to trial. Stimulus presentation lasted 2 s, within which the contrast was modulated between 0 and 2 according to a Gaussian envelope (s.d. = 1/5); maximum contrast was thus reached 1 s after stimulus onset. Apart from the 5.28° patch in which the Gabor was embedded, the remaining field was dark, in order to minimize pupil constriction, which might constrain the effective visual field [8].

### (c) Procedure

To secure a measure of test–retest reliability, we asked all participants to attend for two sessions, at an interval of at least

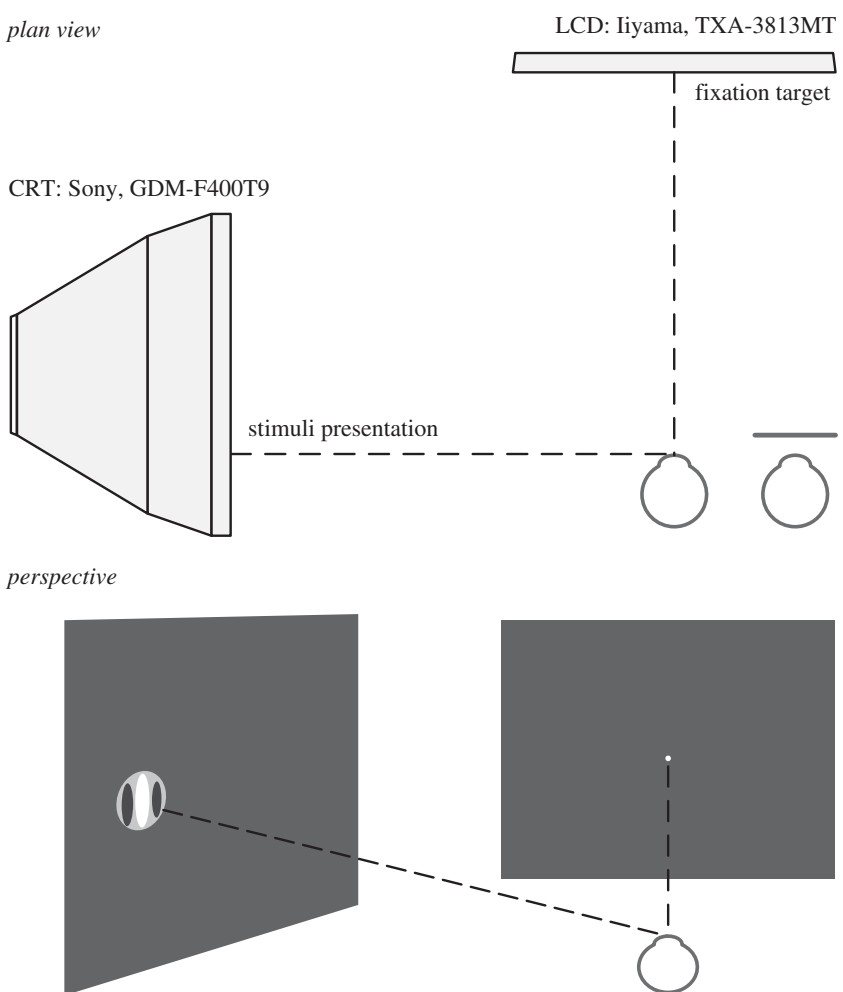

**Figure 1.** Plan view of the experimental arrangements (above) and perspective view of the stimuli (below).

5 days (mean interval = 16.3 days; s.d. = 20.2). In each session, participants started with a training block to familiarize themselves with the task. Six test blocks followed, with brief breaks between blocks as necessary.

A non-speeded two-alternative forced-choice task was used: participants judged the drift direction of the stimulus. Stimulus eccentricity was manipulated by varying the position of the fixation point (giving eccentricities in the range of 75–105°), according to a staircase procedure.

Participants were instructed to keep their gaze on the fixation point throughout the experiment. Each trial began with a 2 s foreperiod and then a double click indicated the onset of the stimulus. The drift direction of the stimulus was random from trial to trial. Participants were free to respond at any time following stimulus onset and the response was acknowledged by a single click. If no response was given in 20 s, the trial was repeated, but participants were encouraged to guess when they were unsure about the right answer. Participants indicated their responses by pressing one of two buttons: the up and down arrow keys on a regular keyboard, corresponding to stimulus directions forward and backward (for a horizontally moving stimulus at an eccentricity of 90°, this is an intuitively comfortable mapping for the participant.) The response triggered the fixation point for the next trial.

In the first trial of the first block of each session, the fixation point was presented at its leftmost position on the fixation screen (resulting in a stimulus eccentricity of 75°). Stimulus eccentricity increased after each pair of consecutive correct responses and decreased after each single incorrect response, a procedure that tracks a performance of 70.7% correct [24]. Each block lasted until the 11th reversal in performance (i.e. when a series of correct responses are followed by an incorrect response or vice versa). Step size was 1° until the third reversal of each block and 0.5° afterwards. From the second block onwards in each session, the first trial's eccentricity matched the eccentricity of the second reversal of the previous block.

## (d) Analysis

The standard deviation of horizontal gaze position (averaged across eyes) was assessed for each session and for each participant, over all times when steady fixation was required. Participants were excluded from analysis if this value exceeded 3° in a session.

For each block, threshold stimulus eccentricity was defined as the mean eccentricity of reversals from the 4th to the 11th (last) reversal in the block. Block means were then further averaged to attain session means for each participant and session. Similarly, standard deviations of block means were calculated for each session.

Test–retest reliability was examined by calculating the Pearson correlation coefficient of session means. The difference between first and second sessions was evaluated by performing paired-sample two-tailed t-tests on session means and standard deviations.

Proc. R. Soc. B **287**: 20200607

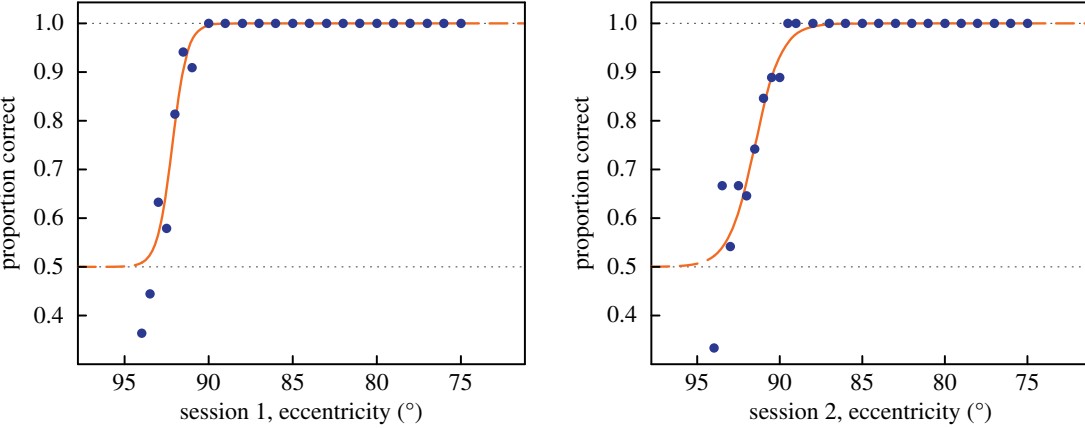

**Figure 2.** Examples of psychometric functions, showing the probability of correct responses as a function of decreasing eccentricity. Data are from the first and second runs for the same participant. (Online version in colour.)

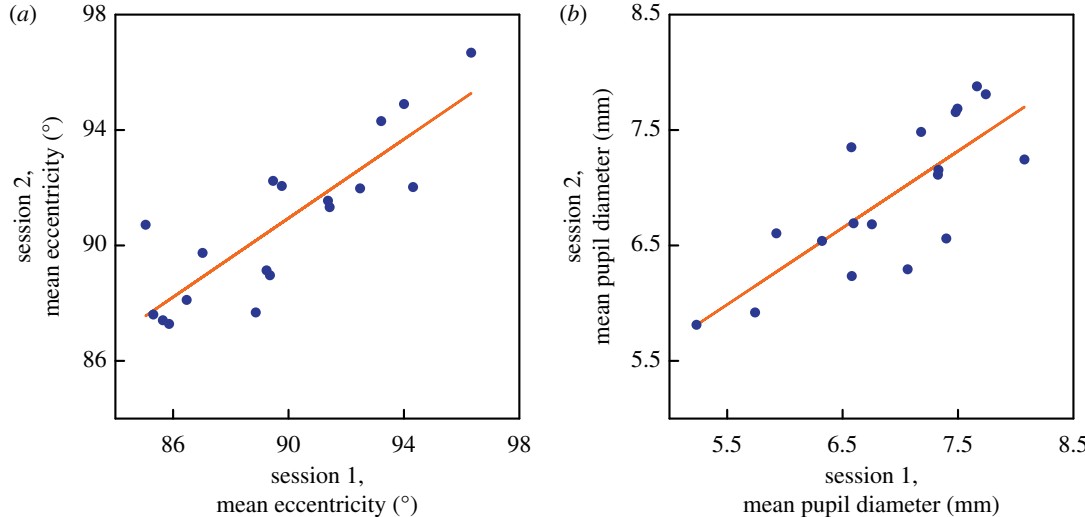

**Figure 3.** (*a*) Test–retest reliability of the mean threshold eccentricity. The *x* and *y* axes show visual angle in degrees. The data of individual participants are plotted (blue points) as session means for the two experimental sessions. Line of best fit plotted in orange. (*b*) Test–retest reliability of measurements of pupil size. The *x* and *y* axes show pupil size in millimetres averaged across eyes and across one session. Data of individual participants are plotted (solid points) as session means for the two experimental sessions. Line of best fit shown by solid line. (Online version in colour.)

## 3. Experiment 1: results

Two participants were excluded owing to excessive eye movements. The standard deviations of their horizontal eye positions were 3.65° and 4.52°. The analyses below are based on the remaining 18 participants. The mean standard deviation for the latter participants was 0.86° and the standard deviation of this value was 0.47°.

Figure 2 illustrates the typical precision of the estimates of the threshold eccentricity. The two psychometric curves (logistic functions fitted using Psignifit v. 2.5.6 [25]) are for the same participant in different sessions and they show how the probability of a correct response varies with the eccentricity of the target patch. The probabilities are pooled across blocks of trials and are cumulated from the eccentricities visited by the adaptive staircase programme, which serves to concentrate test trials in the vicinity of the threshold.

Figure 3*a* shows the relationship between the threshold eccentricities for individual participants in the first and second sessions. Among these healthy participants, there are large individual differences in threshold eccentricity: the range is approximately 10°, from approximately 86° to

approximately 96°. The test–retest reliability is high: thresholds for the two sessions show a Pearson correlation coefficient $r_{16} = 0.85$ ($p < 0.001$).

The mean threshold eccentricity for the first session was 89.73° and that for the second was 90.76°. This difference is small—1°—but is significant ($t_{17} = 2.41$, $p = 0.027$). The standard deviations for the two sessions did not differ significantly (s.d._{1st} = 0.96°, s.d._{2nd} = 1.02°, $t_{17} = 0.28$, $p = 0.78$).

Our gaze-monitor provided concurrent measurements of pupil size, and we have analysed these data for individual observers, since pupil size might be expected to affect psychophysical performance, either by vignetting the optical image or by limiting the absolute light level reaching the far periphery. Our pupil measurements proved to have high test–retest reliability. Figure 3*b* shows the relationship of the mean pupil size (averaged across eyes) for the first and the second sessions. The correlation is high: $r_{16} = 0.781$, $p < 0.001$; and there was no significant difference between sessions: $t_{17} = -0.13$, $p = 0.899$. Nevertheless, although threshold eccentricity and pupil size both prove to be reliable measures, the correlation between them is small and non-significant: $r_{16} = 0.213$, $p = 0.397$.

In this cohort of healthy young adults, there was no significant relationship between age and threshold eccentricity ($r_{16} = -0.32$, $p = 0.20$).

## 4. Experiment 1: discussion

### (a) Estimating the margin of the field

The proposed protocol appears to meet the requirements that are needed for measuring the margin of the visual field. The phase-shifting Gabor stimulus offers a satisfactory compromise between localization and motion. The task is brief, is readily understood, and is comfortable for the participant. Since he or she is asked to make a forced choice of motion direction (rather than a criterion-dependent judgement of presence or absence), it is easy to maintain a consistent response. The use of a fixed Gaussian envelope for the Gabor and the use of a random starting phase for the grating component mean that it is difficult for the participant to rely on secondary cues such as total light flux or scattered light.

By means of these arrangements, rather sharp cut-offs are obtained at the margin of the field (figure 2); and it is worth noting that the slopes of these psychometric functions will reflect the precision of the participant's fixation as well as any actual gradient in the margin itself.

The average improvement between sessions is also very small—of the order of 1°—in contrast with the large learning effects sometimes reported for conventional perimetry [26]. This finding and the high test–retest reliability of the method (figure 3a) suggest that the protocol would be valuable not only to measure individual differences but also to monitor impairments or improvements that occur in disease progression or in the course of therapy.

### (b) Pupil size

It is curious that we did not find a significant correlation between threshold eccentricity and pupil size, though the two measures individually show high test–retest reliability. A relationship might have been expected, if only because the size of the pupil affects the level of illumination reaching the far periphery of the retina [13,14,19].

## 5. Experiment 2: introduction

To exploit further the method used in Experiment 1, we carried out two supplementary experiments with smaller numbers of practiced observers. In Experiment 2, to explore whether we were measuring an anatomical or a functional limit, we measured the threshold stimulus eccentricity at a range of different stimulus contrasts. This experiment was also a necessary preparation for Experiment 3, where we used a silent-substitution procedure to minimize the modulation of the rods and so we needed to know the effect of reducing the effective contrast that was then visible to the cones.

## 6. Experiment 2: methods

Experiments 2 and 3 were carried out with an apparatus that was distinct from, but functionally equivalent to, that of Experiment 1. Stimuli were displayed on a Sony GDM-F400T9 19″ monitor, operated at 85 Hz with a spatial resolution of 1600 × 1200. This was placed to the observer's left, at 600 mm from the eye. A mask of black card was mounted around the stimulus to prevent the observer being aware of reflections or scattering within the front glass layers of the CRT. The individual guns of the monitor were γ corrected using a 'ColorCal 2' photodiode (Cambridge Research Systems) and the spectral power distribution of each gun was measured with a 'Specbos 1201' spectroradiometer (Jeti Technische Instrumente GmbH, Jena). The secondary (fixation) display was an AG Neovo P-19 19-inch monitor placed at 550 mm from the participant and operated at 60 Hz with a 1280 × 1024 resolution. In both supplementary experiments, the average (grey) luminance of the stimulus was 5 cd m$^{-2}$, and thus a 100% contrast stimulus would traverse between 0 and 10 cd m$^{-2}$.

A revised procedure was used to monitor gaze in the supplementary experiments. A Tobii 4c eye-tracker was placed at the base of the fixation monitor and was calibrated using the manufacturer's software. To avoid calibration drift over multiple blocks of trials in the experiment, we introduced a further calibration routine. During this routine, a dot was presented at one of 11 known positions, equally spaced across the horizontal meridian of the monitor. Participants were instructed to press a button when they were centrally fixating the dot. Their eye positions were recorded, and the process was repeated until each of the 11 positions had been sampled 3 times. Then, for each position, a correction was carried out such that the eye-tracker reading closest to the true horizontal location of the dot was chosen. After the calibration, which was carried out before every block, the user was allowed to take a short (less than 30 s) break.

In Experiments 2 and 3, to reduce the need to eliminate data retrospectively, online monitoring of gaze position was used to discourage lapses of fixation as they happened. If eye position deviated from the fixation point by more than ±1.5° of visual angle during the central second of the 2 s presentation, then an aversive tone was played, and the fixation point flashed. The trial was discarded and repeated, with the new direction of the stimulus for the repeat trial being chosen at random.

In other respects, the methods of Experiment 2 were similar to those of Experiment 1. The same adaptive procedure was used to measure two-alternative forced-choice thresholds for discriminating direction of motion, but now thresholds were determined for 10 contrast levels of the Gabor stimulus (10–100%, at intervals of 10%). The order of testing contrast levels was random. The entire procedure was repeated three times and the mean threshold was estimated for each level of contrast. The observers were three of the authors and were all male.

## 7. Experiment 2: results and discussion

The results of this experiment are shown in figure 4a, where the x-axis represents eccentricity and the y-axis represents the log contrast of the Gabor. The three observers differ in their threshold eccentricities, but in each case, the function has a very similar form: over a large range of moderate to high contrasts, the data could be described by a straight line on these semi-log axes, i.e. by a simple exponential function, and then there is a more rapid reduction in threshold eccentricity at the lowest contrasts. The function fitted to each full set of data points in the figure has the form:

$$y = e^{(a(x-b))} + c,$$

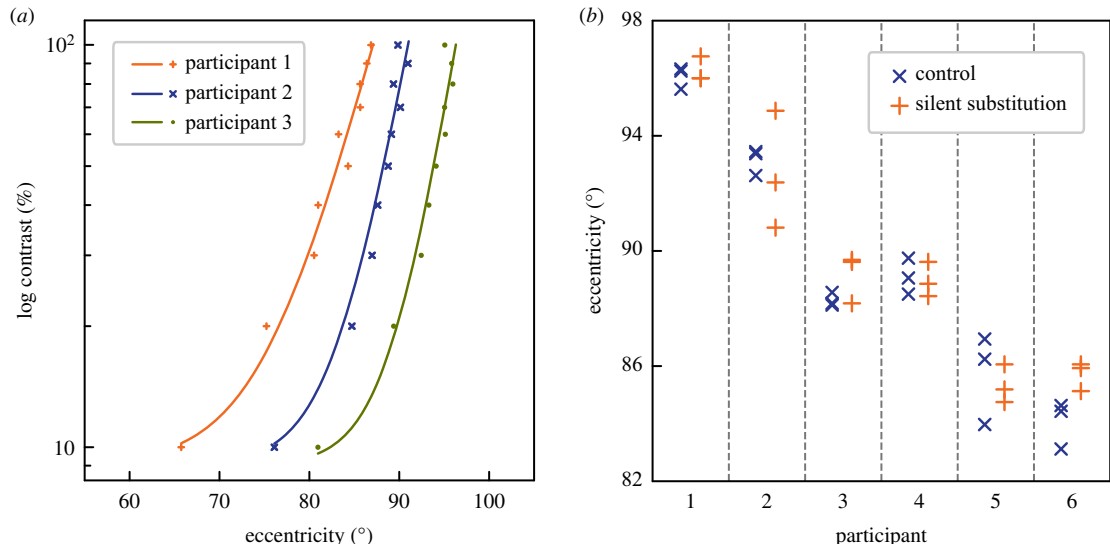

**Figure 4.** (*a*) The change in threshold eccentricity with the contrast of the stimulus. Data are shown for individual observers. The *x*-axis represents the eccentricity of the stimulus at threshold and the *y*-axis represents the log contrast of the Gabor target. For the fitted functions, see text. (*b*) Thresholds for achromatic and for rod-silencing conditions of Experiment 3, shown separately for each observer. Each data point represents the mean threshold eccentricity measured in a single experimental session. (Online version in colour.)

where $y$ is the contrast (%) and $x$ is the eccentricity (°). The constants $a$ and $b$ were set individually for each observer, and correspond, approximately, to the horizontal stretch and the horizontal offset of the exponential function. The constant $c$, which corresponds to the vertical offset of the exponential function, was fixed equal to 9 for all observers.

The steepness of the functions in figure 4*a* suggests (but definitely does not prove) that we are measuring an anatomical limit to the visual field—rather than a functional one that depends substantially on stimulus strength. When the contrast of the Gabor is reduced from 100 to 30%, the threshold eccentricity changes only by an average of 3.9°, rather less than the full width of our Gabor patch. For our third experiment, it is useful to know that the threshold eccentricity is relatively little affected by small variations in the upper range of contrast.

## 8. Experiment 3: introduction

The periphery of the retina is often thought to be dominated by rods, and this assumption has always sat uneasily with the long tradition that the periphery is an organ for detecting motion [27]. For the rods have intrinsically long time-constants, whereas peripheral cones are known to have shorter time constants than even foveal cones [28,29]. In fact, at the edge of the retina, close to the *ora serrata*, cones come to dominate the photoreceptor array, forming a 'cone-rich rim'. The rim was classically described by Schultze [30] and Greeff [31]. A modern study by Williams [32] showed that the cone rim, although it extends all round the anterior margin of the human retina, is most developed on the nasal side (where the temporal field is imaged). Williams estimates that the rim as a whole contains 250 000 cones in comparison to the 75 000 of the fovea. It remains uncertain, however, whether the cone rim is functional [8,15,33].

In the previous experiments, our Gabor stimulus was achromatic. In Experiment 3, the spatial properties of the Gabor were as in the previous experiments, but now we included a condition where the Gabor was formed from counterphase blue and yellow components that were calculated to be of equal scotopic contrast. Owing to chromatic aberration of rays entering the eye very obliquely [8], we cannot be certain that this stimulus achieves a silent substitution for the rods, but we believe that the contrast visible to the rods is now much lower than that visible to the cones. We ask whether the attenuation of rod contrast has a marked effect on the threshold eccentricity.

## 9. Experiment 3: methods

In the rod-silencing Gabor, the chromaticity of any pixel in the output stimulus was now defined by two scalar values, each of which ranged from 0 to 1. The first of these corresponded to the luminance of the blue component, as a proportion of the maximal luminance from the blue gun of the monitor. The yellow component, formed from the red and green guns, was set to have a maximal scotopic luminance equal to the maximum available from the blue gun. For the CIE 10° Observer [34], the Gabor had a *photopic* contrast of 59.11%, while its CIE scotopic contrast was nominally zero. In the control condition, we used an achromatic Gabor of 59.11% contrast. The two conditions were alternated, with the rod-silencing condition first; and each was repeated three times. The procedures for measuring thresholds were as for Experiment 2.

There were six observers (two female). In all but one case, they were members of the laboratory.

## 10. Experiment 3: results and discussion

Figure 4*b* shows, separately for each observer, thresholds for the achromatic and the rod-silencing conditions. Each data point represents the mean eccentricity threshold obtained in a single experimental session. There are systematic individual differences in the threshold eccentricity, but there is no systematic difference between the achromatic condition in which both rods and cones were modulated and the chromatic condition where the modulation of the rods was attenuated. A repeated-measures ANOVA gave no significant difference

between the two thresholds ($F = 0.9735$, d.f. = 1, $p = 0.343$). In fact, the mean difference between the two thresholds is 0.29°, with the rod-silencing condition giving the *greater* threshold eccentricity

The implication is that observers depend on cones, rather than rods, to detect moving stimuli in the far periphery, a conclusion compatible with the histology of this retinal region [32] and with the short time constants of peripheral cones [28].

## 11. General discussion

The method adopted in this study gives an unexpectedly precise estimate of the position of the temporal edge of the field of vision. There are reliable differences between observers in the threshold eccentricity. Reductions in stimulus contrast (Experiment 2) reduce the threshold eccentricity by a few degrees. Contrary to the idea that rod photoreceptors dominate the periphery, rod signals do not seem critical to detecting direction of motion in this region of the field (Experiment 3).

Since the task would be easy to grasp for untrained patients, a protocol of the present kind might be of clinical use—in monitoring changes of the visual field with age or in progressive conditions such as glaucoma [23] or in assessing the effects of prophylactic cryotherapy to prevent retinal detachments that originate from giant retinal tears in type 1 Stickler syndrome [35,36].

It is impressive that human observers can discriminate stimuli that fall at 90° from the line of sight, but the threshold eccentricities reported here are somewhat lower than some values reported in the literature [13], including values from this laboratory [8]. We believe that two factors are critical here:

(i) Previous studies have not monitored eye movements during the measurements. Even practiced observers are often unaware of lapses of fixation and of saccades in the direction of a suddenly appearing peripheral target.

(ii) It may be valuable to make a distinction between the limit of image-forming vision and the limit for simple detection of light. It is quite possible that in the laboratory, and in the real world, observers can detect light reflected or scattered from anterior surfaces of the eye, or indeed from the eye lashes; and that such cues allow detection of stimuli beyond 100° from the line of sight. But the present measurements, where cues from scattered light were minimized and where fixation was monitored, probably represent the limits to image-forming vision.

Our working hypothesis is that a single type of neural channel—the 'last channel'—survives in the farthest periphery of the visual field. We propose that this channel carries directional information and responds to low spatial frequencies. Our psychophysical results cannot determine whether the channel arises in the retina itself or more centrally. It is often held that the visual analysis of motion, despite its importance in guiding several aspects of human behaviour, is 'encephalized' in man. In the retinae of mice and of rabbits, directionally selective ganglion cells have been extensively studied (e.g. [37,38]), but the presence of such cells in primates has been doubted or denied. To *et al.* [6], however, noted that the enhanced psychophysical sensitivity to four, near-cardinal, directions, observed in the far temporal field, did recall the presence of four preferred directions for ON–OFF directionally selective ganglion cells in the rabbit retina [39]. Dacey and co-workers [40,41] have now shown that the recursive bistratified ganglion cell of the macaque retina is in fact directionally selective. So far, however, there appears to be only a single mosaic of these cells in primates, suggesting that multiple directional preferences are not present at a given retinal location. Nevertheless, our provisional hypothesis is that the sparse ganglion cells described by Polyak in the far periphery of the retina are either themselves directionally selective—or have the properties needed to supply a directionally selective channel that arises more centrally.

Ethics. Written informed consent was given before taking part, and all procedures conformed to the Declaration of Helsinki. Ethics permission was given by the Psychology Research Ethics Committee, Cambridge University (PRE.2018.078).

Data accessibility. The data supporting these experiments can be obtained from Veto, Peter *et al.* (2020), 'The last channel': Vision at the temporal margin of the field, Dryad, Dataset, https://doi.org/10.5061/dryad.v6wwpzgs5.

Authors' contributions. P.B.M.T., P.A., and J.D.M. designed the project and prepared the funding application. P.V. prepared and carried out Experiment 1. T.W. prepared and carried out Experiments 2 and 3. All authors contributed to the preparation of the text.

Competing interests. We declare we have no competing interests.

Funding. Supported by a grant from The Evelyn Trust (17/18) and by BBSRC (BB/S000623/1). P.B.M.T. was supported by the National Institute for Health Research (NIHR) Biomedical Research Centre at Moorfields Eye Hospital NHS Foundation Trust and UCL Institute of Ophthalmology.

Acknowledgements. We thank Dr Chie Takahashi for preparing figure 1.

Disclaimer. The views expressed are those of the authors and not necessarily those of the NHS, NIHR, or Department of Health.

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
