## [Reviewer comments · Proceedings of the Royal Society B: Biological Sciences]

Review History

RSPB-2020-0607.R0 (Original submission)

Review form: Reviewer 1 (Jochen Laubrock)

Recommendation

Accept with minor revision (please list in comments)

Scientific importance: Is the manuscript an original and important contribution to its field?

Good

General interest: Is the paper of sufficient general interest?

Good

Quality of the paper: Is the overall quality of the paper suitable?

Excellent

Is the length of the paper justified?

Yes

Should the paper be seen by a specialist statistical reviewer?

No

Do you have any concerns about statistical analyses in this paper? If so, please specify them explicitly in your report.

No

It is a condition of publication that authors make their supporting data, code and materials available - either as supplementary material or hosted in an external repository. Please rate, if applicable, the supporting data on the following criteria.

Is it accessible?

N/A

Is it clear?

N/A

Is it adequate?

N/A

Do you have any ethical concerns with this paper?

No

Comments to the Author

I enjoyed reading this well-written paper. Please see attached comments for details (See Appendix A).

Review form: Reviewer 2

Recommendation

Accept with minor revision (please list in comments)

Scientific importance: Is the manuscript an original and important contribution to its field?

Excellent

General interest: Is the paper of sufficient general interest?

Good

Quality of the paper: Is the overall quality of the paper suitable?

Excellent

Is the length of the paper justified?

Yes

Should the paper be seen by a specialist statistical reviewer?

No

Do you have any concerns about statistical analyses in this paper? If so, please specify them explicitly in your report.

No

It is a condition of publication that authors make their supporting data, code and materials available - either as supplementary material or hosted in an external repository. Please rate, if applicable, the supporting data on the following criteria.

Is it accessible?

Yes

Is it clear?

Yes

Is it adequate?

Yes

Do you have any ethical concerns with this paper?

No

Comments to the Author

This is a nice old-fashioned psychophysics study that provides new and very useful information about the function of the far temporal periphery. They develop a way to measure the limiting eccentricity in humans, and show there are large but reliable differences between observers. The estimates are robust to contrast reduction, suggesting that the limit is probably anatomical. The experiments have been performed and analyzed with scientific rigor. It is difficult to make new basic discoveries about visual function these days: this is definitely appropriate for the Proceedings. I have no major criticisms, only some comments that may or may not be useful.

Personally, I think the paper is too short and compact to divide into three different experiments (which are in fact all variants of the same). The methods etc are sufficiently similar to have a single methods, results and discussion sections, with sub-headings. But this is the authors' choice.

I am unclear what the pupillometry adds. Perhaps this can be better justified – or removed. If they do choose to keep it, perhaps show the lack of correlation between eccentricity and pupil size.

The test-retest correlations are impressive, and important. Would it make sense to calculate Cronbach's alpha for them? – just an idea, not at all sure it will help.

Surely Figure 4B would work better plotting rod-isolated results against controls, with different symbols for each participant? They should hug the equality line.

In general the graphs could be prettier. Maybe put the data into a better plotting program, such as Origin.

In the introduction you mention the prostriata in humans. I was hoping that may be followed up in discussion. Do you have any thoughts?

All up, nice work, congratulations!

David Burr
Perth 30/3/2020

Decision letter (RSPB-2020-0607.R0)

01-Apr-2020

Dear Dr Mollon

I am pleased to inform you that your Review manuscript RSPB-2020-0607 entitled "The last channel': Vision at the temporal margin of the field" has been accepted for publication in Proceedings B.

The referee(s) do not recommend any further changes. Therefore, please proof-read your manuscript carefully and upload your final files for publication. Because the schedule for publication is very tight, it is a condition of publication that you submit the revised version of your manuscript within 7 days. If you do not think you will be able to meet this date please let me know immediately.

To upload your manuscript, log into <http://mc.manuscriptcentral.com/prsb> and enter your Author Centre, where you will find your manuscript title listed under "Manuscripts with Decisions." Under "Actions," click on "Create a Revision." Your manuscript number has been appended to denote a revision.

You will be unable to make your revisions on the originally submitted version of the manuscript. Instead, upload a new version through your Author Centre.

1) A text file of the manuscript (doc, txt, rtf or tex), including the references, tables (including captions) and figure captions. Please remove any tracked changes from the text before submission. PDF files are not an accepted format for the "Main Document".

2) A separate electronic file of each figure (tiff, EPS or print-quality PDF preferred). The format should be produced directly from original creation package, or original software format. Please note that PowerPoint files are not accepted.

3) Electronic supplementary material: this should be contained in a separate file from the main text and the file name should contain the author's name and journal name, e.g. `authorname_procb_ESM_figures.pdf`

All supplementary materials accompanying an accepted article will be treated as in their final form. They will be published alongside the paper on the journal website and posted on the online figshare repository. Files on figshare will be made available approximately one week before the accompanying article so that the supplementary material can be attributed a unique DOI. Please see: <https://royalsociety.org/journals/authors/author-guidelines/>

4) Data-Sharing and data citation

It is a condition of publication that data supporting your paper are made available. Data should be made available either in the electronic supplementary material or through an appropriate repository. Details of how to access data should be included in your paper. Please see <https://royalsociety.org/journals/ethics-policies/data-sharing-mining/> for more details.

<http://datadryad.org/submit?journalID=RSPB&manu=RSPB-2020-0607> which will take you to your unique entry in the Dryad repository.

Once again, thank you for submitting your manuscript to Proceedings B and I look forward to receiving your final version. If you have any questions at all, please do not hesitate to get in touch.

Sincerely,
Dr Robert Barton
mailto:proceedingsb@royalsociety.org

Associate Editor Board Member: 1

Comments to Author:

We have now heard from two of our experts. Both were positive about your manuscript -- but had some comments and suggestions. I am pleased to recommend acceptance of your manuscript after you have dealt with the points they have raised.

Reviewer(s)' Comments to Author:

Referee: 1

Comments to the Author(s)

I enjoyed reading this well-written paper. Please see attached comments for details.

Referee: 2

Comments to the Author(s)

This is a nice old-fashioned psychophysics study that provides new and very useful information about the function of the far temporal periphery. They develop a way to measure the limiting eccentricity in humans, and show there are large but reliable differences between observers. The estimates are robust to contrast reduction, suggesting that the limit is probably anatomical. The experiments have been performed and analyzed with scientific rigor. It is difficult to make new basic discoveries about visual function these days: this is definitely appropriate for the Proceedings. I have no major criticisms, only some comments that may or may not be useful.

Personally, I think the paper is too short and compact to divide into three different experiments (which are in fact all variants of the same). The methods etc are sufficiently similar to have a single methods, results and discussion sections, with sub-headings. But this is the authors' choice.

I am unclear what the pupillometry adds. Perhaps this can be better justified – or removed. If they do choose to keep it, perhaps show the lack of correlation between eccentricity and pupil size.

The test-retest correlations are impressive, and important. Would it make sense to calculate Cronbach's alpha for them? – just an idea, not at all sure it will help.

Surely Figure 4B would work better plotting rod-isolated results against controls, with different symbols for each participant? They should hug the equality line.

In general the graphs could be prettier. Maybe put the data into a better plotting program, such as Origin.

In the introduction you mention the prostriata in humans. I was hoping that may be followed up in discussion. Do you have any thoughts?

All up, nice work, congratulations!
David Burr
Perth 30/3/2020

Author's Response to Decision Letter for (RSPB-2020-0607.R0)

See Appendix B.

Decision letter (RSPB-2020-0607.R1)

14-Apr-2020

Dear Dr Mollon

I am pleased to inform you that your manuscript entitled "The last channel': Vision at the temporal margin of the field" has been accepted for publication in Proceedings B.

Your article has been estimated as being 8 pages long. Our Production Office will be able to confirm the exact length at proof stage.

Open Access

You are invited to opt for Open Access, making your freely available to all as soon as it is ready for publication under a CCBY licence. Our article processing charge for Open Access is £1700. Corresponding authors from member institutions (<http://royalsocietypublishing.org/site/librarians/allmembers.xhtml>) receive a 25% discount to these charges. For more information please visit <http://royalsocietypublishing.org/open-access>.

Paper charges

Sincerely,
Proceedings B
<mailto:proceedingsb@royalsociety.org>

Appendix A

Review of RSPB-2020-0607 'The last channel': Vision at the temporal margin of the field

The paper reports three experiments aiming to functionally characterize vision in the farthest periphery of the temporal visual field. Results suggest that extreme peripheral vision is directionally sensitive to temporally modulated stimuli of very low spatial frequency. One function of such a filter may be to rapidly alert us to sudden movement in the periphery. Together with anatomical evidence--relatively high cone density along the nasal retinal margin, where the extreme lateral periphery of the visual field is projected, and a short time constant of cones--the present result that psychophysically luminance contrasts do not appear to be critical for performance is reasonably interpreted to indicate that this channel is driven by cone vision. The merits of the paper are (a) to develop a novel method to reliably measure this kind of vision, and to show that (b) there are stable individual differences therein, and (c) it is unlikely to be performed by rods.

I enjoyed reading this well-written paper. The results are novel, and the psychophysical experiments appear to be well-designed and carefully conducted (I liked the idea of moving the fixation point rather than the peripheral stimulus, for example). These are possibly the first experimental demonstrations of the alerting function of the foveal rim that has been speculated about for long (e.g., Williams, 1991). I only have a few comments.

Major points:

- ll. 190-195, Figure 2: If this is the typical performance, I wonder why proportion correct is consistently dropping below the guessing level at near eccentricities. Do the authors have any speculation about it? Or could it be that just something is wrong with the y-axis labels, or maybe the fitting procedure would need to be adjusted to the number of response alternatives?
- The concept of a channel features prominently, even in the title. It might help to educate the more general reader what a "channel" means to the vision scientist; perhaps a filter tuned to some specific type of information? Indeed this is done in the first two sentences, but it wouldn't hurt to paraphrase it again later, maybe in the discussion.

Minor points:

- If the optimal stimulus for the far periphery a moving stimulus of 0.3 cpd (ll. 77-78), why did the authors choose a grating of 0.55 cpd (l. 136)? This might be another factor to consider in why the threshold eccentricities reported here are somewhat lower than some values reported in the literature (ll. 398ff.).
- (ll. 202) " $p < 0.0001$ ": maybe better $p < 0.001$?
- I was a bit puzzled to read about retest reliability for pupil size measurement, which felt out of context, or distracting from the main message. I realize that it might serve as a comparison, and indeed the

zero correlation of pupil size with threshold eccentricity is interesting. So the paragraph should just be contextualized better.

- Contrast is typically varied on a logarithmic scale, whereas it seems to have been varied more or less linearly here, so that lower levels of contrasts are relatively undersampled. If the ordinate in Figure 4A were plotted logarithmically, wouldn't the curves look more or less linear? I wonder how this affects the description of the results that "at moderate to high contrasts the threshold eccentricity varies only slightly as contrast is reduced and then there is a more rapid reduction of threshold eccentricity at low contrasts" and the interpretation that "we are measuring an anatomical limit to the visual field" rather than a functional one that depends substantially on stimulus strength" (which I didn't fully understand anyhow). In case the function still looks upward concave, what would the relatively low threshold eccentricity at lower levels of contrast mean physiologically?

- A parametric exponential function was fitted to the data in Figure 4A. Why was this functional form chosen? Do the parameters a , b , c , and d have any interpretation? (Just wondering)

- (ll. 349-359) Was gamma correction used? I am no expert in color vision, and I believe that the procedure used is a good first attempt at attenuating rod vision, but to really conclude that rods were silenced wouldn't it be better to calibrate the colors individually using heterochromatic flicker or minimum motion photometry?

Jochen Laubrock

Appendix B

Response to Reviewers

We are very grateful to the reviewers for their kind remarks on the paper and their positive and helpful suggestions. We have acted on many of their suggestions and have made several local changes in the ms:

Reviewer 1.

- ll. 190-195, Figure 2: If this is the typical performance, I wonder why proportion correct is consistently dropping below the guessing level at near eccentricities. Do the authors have any speculation about it? Or could it be that just something is wrong with the y-axis labels, or maybe the fitting procedure would need to be adjusted to the number of response alternatives?

This is a technical by-product of the adaptive, 'staircase', procedures widely used in psychophysics. The particular procedure we used (go up after 1 error, down after 2 correct) means that the procedure tracks 71% correct. So probabilities in this region of the function are the most precise. But extreme eccentricities (i.e. positions at the left of the abscissa) are visited rarely and so the probabilities are based on tiny samples. Moreover, the rules of the staircase will tend to force the probabilities below 50%. Consider the most extreme eccentricity that is visited by a given observer and where he or she is simply guessing. On half the visits, the response will be incorrect ($p = 0.5$) and the staircase returns to an easier level. On the other half of trials there will be a mixture of correct and incorrect responses, with the staircase returning to an easier level as soon as an incorrect response occurs. So the *average* probability will be below 0.5. It would take a disproportionate insertion in our main text to set out this interesting but very technical point and, being close to the word limit, we have hesitated to do so.

- The concept of a channel features prominently, even in the title. It might help to educate the more general reader what a "channel" means to the vision scientist; perhaps a filter tuned to some specific type of information? Indeed this is done in the first two sentences, but it wouldn't hurt to paraphrase it again later, maybe in the discussion.

At the last paragraph of the Discussion, we have replaced 'channel' by 'neural pathway'.

- If the optimal stimulus for the far periphery a moving stimulus of 0.3 cpd (ll. 77-78), why did the authors choose a grating of 0.55 cpd (l. 136)? This might be another factor to consider in why the threshold eccentricities reported here are somewhat lower than some values reported in the literature (ll. 398ff.).

In the Methods, we now explicitly explain our choice of stimulus frequency.

- (ll. 202) " $p < 0.0001$ ": maybe better $p < 0.001$?

We have made this change throughout.

- I was a bit puzzled to read about retest reliability for pupil size measurement, which felt out of context, or distracting from the main message. I realize that it might serve as a comparison, and indeed the zero correlation of pupil size with threshold eccentricity is interesting. So the paragraph should just be contextualized better.

In the Results for Experiment 1, we now explicitly give the reasons for this analysis.

- Contrast is typically varied on a logarithmic scale, whereas it seems to have been varied more or less linearly here, so that lower levels of contrasts are relatively undersampled. If the ordinate in Figure 4A were plotted logarithmically, wouldn't the curves look more or less linear? I wonder how this affects the description of the

results that "at moderate to high contrasts the threshold eccentricity varies only slightly as contrast is reduced and then there is a more rapid reduction of threshold eccentricity at low contrasts" and the interpretation that "we are measuring an anatomical limit to the visual field – rather than a functional one that depends substantially on stimulus strength" (which I didn't fully understand anyhow). In case the function still looks upward concave, what would the relatively low threshold eccentricity at lower levels of contrast mean physiologically?

We have now plotted log contrast in Figure 4 (a), as the reviewer asks, and we have modified the relevant text.

- (ll. 349-359) Was gamma correction used? I am no expert in color vision, and I believe that the procedure used is a good first attempt at attenuating rod vision, but to really conclude that rods were silenced wouldn't it be better to calibrate the colors individually using heterochromatic flicker or minimum motion photometry?

Yes, gamma correction was used, and we now make this explicit. In addition, in Experiment 3, the spectral power distributions were measured with a spectroradiometer. Photometric corrections for individual observers were not attempted, since our main purpose was to achieve an approximate equation of *rod* responses and there is rather little variation between observers in the spectral sensitivity of the rods. In addition Experiment 2 shows that small errors in the estimated *cone* contrasts are likely to have little effect on performance.

Reviewer 2.

In response to the suggestions of Reviewer 2 we have inserted a justification of the inclusion of the measurements of pupil size (see above).

We tried the alternative way of plotting Figure 4B but in fact it did not make the point so clearly, and so we have retained the original.

We have redone all the figures using Adobe Illustrator and hope that they are now all improved.